# The In Situ Hydrothermal and Microwave Syntheses of Zinc Oxides for Functional Cement Composites

**DOI:** 10.3390/ma15031069

**Published:** 2022-01-29

**Authors:** Izabela Klapiszewska, Adam Kubiak, Anna Parus, Marcin Janczarek, Agnieszka Ślosarczyk

**Affiliations:** 1Institute of Building Engineering, Faculty of Civil and Transport Engineering, Poznan University of Technology, Piotrowo 3, PL-60965 Poznan, Poland; izabela.klapiszewska@put.poznan.pl; 2Institute of Chemistry and Technical Electrochemistry, Faculty of Chemical Technology, Poznan University of Technology, Berdychowo 4, PL-60965 Poznan, Poland; adam.kubiak@put.poznan.pl; 3Institute of Chemical Technology and Engineering, Faculty of Chemical Technology, Poznan University of Technology, Berdychowo 4, PL-60965 Poznan, Poland; anna.parus@put.poznan.pl (A.P.); marcin.janczarek@put.poznan.pl (M.J.)

**Keywords:** zinc oxide, cement composite, antibacterial test, strength test

## Abstract

This study presents the results of research on cement mortars amended with two zinc oxides obtained by two different methods: hydrothermal ZnO-H and microwave ZnO-M. Our work indicates that, in contrast to spherical ZnO-H, ZnO-M was characterized by a columnar particle habit with a BET surface area of 8 m^2^/g, which was four times higher than that obtained for hydrothermally obtained zinc oxide. In addition, ZnO-M induced much better antimicrobial resistance, which was also reported in cement mortar with this oxide. Both zinc oxides showed very good photocatalytic properties, as demonstrated by the 4-chlorophenol degradation test. The reaction efficiency was high, reaching the level of 90%. However, zinc oxides significantly delayed the cement binder setting: ZnO-H by 430 min and ZnO-M by 380 min. This in turn affected the increments in compressive strength of the produced mortars. No significant change in compressive strength was observed on the first day of setting, while significant changes in the strengths of mortars with both zinc oxides were observed later after 7 and 28 days of hardening. As of these times, the compressive strengths were about 13–15.5% and 12–13% higher than the corresponding values for the reference mortar, respectively, for ZnO-H and ZnO-M. There were no significant changes in plasticity and flexural strength of mortars amended with both zinc oxides.

## 1. Introduction

Nowadays, nanotechnology is a field with a very wide range of applications and a high rate of development. The significant growth of nanotechnology is best evidenced by the number of scientific articles, which for the “nano-technology” keyword is associated with 85,240 documents, 72,408 of which were published in the last 10 years. Year by year, the number of works on this subject is increasing, and in 2021 alone, as many as 11,000 scientific articles in the field of nanotechnology were published [1]. The wide application possibilities of nanoparticles are also successfully used in the construction industry. The authors of many studies have investigated the influence of nanoparticles of various oxides on the properties of cement composites. Examples include, e.g., SiO_2_, Al_2_O_3_, Fe_2_O_3_, ZnO, TiO_2_ and CuO [2,3,4,5,6,7,8] nanoparticles. The purpose of incorporating these materials into the cement matrix is to thicken the structure by increasing its compactness. In addition, nanoparticles are highly reactive because of their usually high BET surface area. The hydration of nanoparticles induces a nucleation effect, accelerating the cement bonding process. Another beneficial effect of the applied nanoparticles is the improvement of the interfacial transition zone between the aggregate and the cement paste [9].

A widely used oxide that has become more important in recent times, zinc oxide (ZnO), due to its desirable properties (both physical and chemical), has found application in many industries. One is the pharmaceutical and cosmetic industry, where it is incorporated as an ingredient in creams, powders, dental pastes, etc. Another industrial sector utilizing ZnO is the rubber industry, where it is used as a filler and activator. In addition, ZnO is exploited in the textile industry as a UV radiation absorber. Other uses for this oxide can be found in electronics and electrical engineering industries, as well as in pollution remedial activities, notably as a photocatalysis [10].

ZnO is also applied for amending cements. Key advantages in harnessing the attributes of this material lay in its universality, durability, low cost and non-toxicity [11]. It is worth noting that both zinc oxide and its compounds or materials containing zinc are characterized by affecting a delay in cement hydration [12,13]. The vast majority of retarders based on materials exploiting zinc’s properties are made of zinc oxide or nano-ZnO [12]. The use of ultra-fine ZnO nanoparticles allows one not only to delay the hydration reaction, but also to reduce the formation of C–H structures. It also increases the density of the microstructure (replacing water molecules present in the void spaces), and improves contact in the interfacial zone, creating a more compact structure with lower permeability [14].

Another advantage of utilizing the attributes of ZnO nanoparticles is related to their photocatalytic properties (as with TiO_2_ particles) [15,16]. Due to the use of photocatalysts, it is possible to counteract global warming by increasing the level of oxygen [16]. Zinc oxide is characterized by one more property, which is currently of significant importance. It is an antibacterial material that displays excellent growth-inhibiting activity against a wide spectrum of pathogens, incl. *Staphylococcus aureus*, *Klebsiella pneumoniae* or *Escherichia coli* [17,18,19]. Research indicates that, as the particle size of the zinc oxide is reduced, the effectiveness of the material increases, resulting in a lowered abundance or complete elimination of bacteria. Therefore, it can be concluded that the antibacterial properties depend to a high extent on the size, shape, morphology, and presence of defects in the crystal structure or the BET surface area of the material [18]. The authors of the article [18] present the antibacterial properties of two types of zinc oxide, synthesized using diverse methods. The ZnO particles were obtained by way of the precipitation method and the disperser-assisted sonochemical method. The resulting materials were characterized by antibacterial properties against Gram-positive and Gram-negative microorganisms. A comparison of both methods of synthesis showed that more favorable properties were achieved for particles obtained with the sonochemical method. The produced material was characterized by an easy-to-control morphology, high thermal stability, small-sized particles, good crystallinity and considerable-size BET surface area. Overall, the antibacterial properties are now being extensively harnessed, mainly for water purification in the presence of *E. coli* [20].

Zinc oxide can be obtained by various methods. These include mechano-chemical processes, precipitation processes, sol-gel, emulsion and microemulsion, hydrothermal and microwave [10]. The authors of this paper have focused their attention on the preparation of zinc oxide by microwave and hydrothermal methods. Microwave synthesis saves energy and achieves a higher reaction rate, while heating is carried out volumetrically. This distributes the heat evenly and results in larger quantities of product obtained, as compared to conventional heating methods [21]. Furthermore, in their work, Krishnakumar et al. [21], using zinc nitrate (ZnO(NO_3_)_2_·6H_2_O) as a precursor, obtained flower-shaped crystalline ZnO structures. In their research, the whole microwave treatment process took only 10 min, and hydrazine hydrate solution was used to achieve the correct pH = 8.

Another synthesis method employing the microwave method was proposed by Kubiak et al. [22]. The researchers used zinc acetate dihydrate and triethylamine to achieve the desired pH (8, 10 or 12). Microwave treatment lasted 5 min, at 120 °C with 300 W irradiation power. The obtained zinc oxide was characterized by a well-formed wurtzite phase from all pH environments used. However, depending on the pH, ZnO with different shapes was obtained. Samples from pH 8 and 10 environments were characterized by hexagonal and pyramidal shapes, while for pH 12, a flower-like structure was obtained.

Another group of researchers [23] performed the synthesis of ZnO nanoparticles from zinc nitrate-6-hydrate, zinc acetate dehydrate, and hydrazine hydrate and ammonia solution. They conducted the synthesis at pH~11.5, applying two different irradiation groups: 15 min at 510 W and 10 min at 680 W. The obtained nanoparticles after being subjected to lower irradiation power yielded needle-shaped particles, while for higher irradiation powers, the yielded ZnO displayed a flower-shaped morphology.

In contrast, zinc oxide prepared by the hydrothermal method involves using high heat and pressure (100–1000 °C and 1–10,000 atm) with water as a solvent in an autoclave. The hydrothermal method is attractive for its simplicity and environmentally friendly conditions [24]. The products obtained by this method are of good quality, but this method has some disadvantages; high energy consumption and expensive equipment (autoclaves) are of necessity [25]. Nevertheless, the hydrothermal synthesis of ZnO nanostructures is simple and efficient, and has recently received considerable attention.

Various precursors have been successfully employed in aqueous media to synthesize ZnO nanostructures with different morphologies, the most popular including a mixture of zinc nitrate and hexamine [24]. In addition to the heating mechanism, other factors such as precursor type, surfactants, alkalinity source, annealing temperature and dopants affect the morphological structure [25]. ZnO nanostructures are attractive candidates for applications in a wide range of devices such as solar cells, sensors, detectors, power generators, and artificial structures for tissue engineering [24].

As part of this study, zinc oxide was obtained using two methods: hydrothermal and microwave. The obtained oxides were subjected to a thorough morphological, microstructural and physicochemical analysis. Additionally, their photocatalytic and antibacterial properties were assessed. The final stage of the work, which, according to the authors’ knowledge, has not yet been tested, was the use of the obtained oxides as functional admixtures for cement composites. For cement mixes prepared with their participation, we assessed the flow rate, the beginning of setting time, and then the bending and compressive strengths, in order to determine their mechanical strength after 1, 7 and 28 days of maturing. Moreover, we examined the microstructure.

## 2. Materials and Methods

### 2.1. Materials

Zinc acetate dihydrate (≥98%, Sigma-Aldrich, St. Louis, MO, USA), and urea (p.a., Sigma-Aldrich) were exploited for the synthesis of ZnO powders. All reagents used required no further purification as they were of analytical purity. The water used during the experiments was deionized.

To produce cement composites, Portland cement CEM I 42.5R from Górażdże Cement Company SA, Górażdże, Poland (that meets the requirements of the EN 197-1 standard) and a standard quartz sand (that meets the requirements of the EN 196-1 standard EN) from Kwarcmix, Tomaszów Mazowiecki, Poland were employed.

### 2.2. Preparation and Characterization of Zinc Oxide

The first step in the synthesis of zinc oxide was the addition of 1 g of urea to 100 mL of a 5% solution of zinc acetate dihydrate, which was stirred further until it was completely dissolved. The obtained mixture was then moved to a hydrothermal reactor (Parr Instrument Co., St, Moline, IL, USA) or microwave reactor (SPD80, CEM Corporation, Smith Farm Road, Matthews, NC, USA). The conditions of hydrothermal and microwave treatment are presented in Table 1. In the next step, when the reactor reached room temperature, the reaction products were filtered and washed three times with water and ethanol. The final stage was drying for 6 h at 60 °C.

The zinc oxide produced was characterized to determine the crystalline structure (X-ray diffraction—XRD), morphology (scanning electron microscopy—SEM, and transmission electron microscopy—TEM), the parameters of the porous structure (low-temperature N_2_ sorption), and the thermal stability (thermogravimetric analysis–TGA, and the first derivative of the TGA curve—the DTG curve). A detailed description of the physicochemical analyses carried out is presented elsewhere [22,26].

### 2.3. Photocatalytic Properties of Zinc Oxide

The degradation of 4-chlorophenol was used to evaluate the photooxidation activity of the produced materials. An LED lamp based on a COB (chip-on-board) system was used as a light source. This incorporated a diode with a wavelength of 395 nm and a power of 50 W (BRIDGELUX, Fremont, CA, USA). Due to the power requirements, it was equipped with an aluminum active radiator (with fan) to dissipate the resulting heat. In the final stage, the obtained system was connected to the controller (TOPXIN Electronics Co., LTD, Shenzhen, China) and the power was obtained according to the previously assumed value.

The photo-oxidation test began with 100 cm^3^ of the investigated material and 100 mg of the photocatalyst being introduced into the reactor. Then, to determine the adsorption/desorption equilibrium, the suspension was homogenized in the dark (30 min). The next step was to irradiate the reaction mixture using a UV-LED lamp. Then, to determine the maximum absorbance of 4-chlorophenol, 3 cm^3^ of the suspension was collected and filtered using a syringe filter (Macherey-Nagel, Duren, Germany). The resulting filtrate was analyzed on a UV-VIS spectrophotometer (V-750, Jasco, Tokyo, Japan). This activity was repeated every 30 min for 180 min, after which exposure was terminated. The maximum absorbance at a wavenumber 280 nm was determined. The calibration curve method was used to determine the photocatalytic activity. The determined curve with the formula y = 0.01x − 0.0277 considers x as the concentration of 4-chlorophenol and y as the maximum value of absorbance.

### 2.4. Production and Characterization of Cement Composites

The preparation of cement composites amended with 0.1 wt.% of zinc oxide consisted of initially dispersing the dry ingredients in 225 mL of mixing water, using a magnetic stirrer (~5 min), and then adding them to the mixer, in which 450 g of cement had previously been placed. The mixer was started at low speed (140 ± 5 min^−1^), and 1350 g of aggregate dosing was conducted after 30 s, which lasted for 30 s. For the next 30 s, the mixer was operated at high speed (285 ± 10 min^−1^). The subsequent step was a 90 s pause, which was followed by 60 s of high-speed mixing. The prepared cement mortar was then introduced into the mold to obtain cement beams with dimensions of 40 mm × 40 mm × 160 mm. The mortar beams were released from the molds after 24 h and stored in water until the test. It should be noted that the mixing method complies with the PN-EN 196-1 standard [27]. The preparation of cement composites is schematically shown in Figure 1, and the composition of cement mixtures is listed in Table 2.

The fresh cement mortar was subjected to a standard consistency test, involving the spread of fresh mortar on a shaking table. The aim of this study was to measure the diameter of the spread in two perpendicular directions after the ten applications of a shaking force. The test was conducted in accordance with the PN-EN 1015-3 standard [28].

Mechanical strength tests were carried out using a MATEST (Treviolo (BG), Italy) testing machine after 1, 7 and 28 days. A flexural strength sample with dimensions of 40 × 40 × 160 mm^3^ was placed in the testing machine and a constant load was applied until breaking. Half of the previously used beam was then placed between two plates, and the compressive strength was subsequently assessed in the suspended portion under constant load. The undertaken compressive strength test complies with PN-EN 196-1 [27].

Additionally, the initial setting time of the cement paste was ascertained on the basis of the PN-EN 196-3 standard [29]. The cement paste was prepared in a mechanical mixer. The time when water and cement were introduced in the mixing bowl was designated as time zero. The mixer was then switched on for 90 s at low speed, followed by a 15 s pause. In the following step, mixing was continued for 90 s, and the prepared paste was subsequently transferred to a lightly oiled Vicat ring placed on a glass plate. The mortar-filled ring was placed under the needle of the Vicat apparatus and the needle was lowered to contact with the surface of the mortar, then the screw was released, allowing it to sink into the hole. The reading was conducted after 30 s or when the needle was fully inserted. The beginning of the setting time was considered to be the time that elapsed from time zero until the distance of the needle from the base plate reached 6 ± 3 mm.

### 2.5. Antimicrobial Properties of Pristine Admixtures and Cement Composites

The ability of the tested materials to inhibit microbial growth was investigated using the serial dilutions method [30,31]. The materials in the form of powders (100 mg) were introduced into vials (2 mL) and suspended in sterile water. Then, the suspensions were homogenized using a shaker and used as a stock solution, which was diluted at a 1:1 ratio using water. As a result, a total of eight suspensions with decreasing concentrations were obtained, which were placed in a refrigerator (4 °C) prior to conducting the tests.

The growth-inhibiting properties were tested towards selected model strains of microorganisms, which included a fungi (*C. albicans*) two Gram (+) bacteria (*B. cereus* and *S. aureus*) and three Gram (−) bacteria (*P. putida*, *P. aeruginosa* and *E. coli*). The microbes were selected due to their widespread occurrence and the fact that most (with the exception of *P. putida*) can be opportunistic human pathogens. In order to conduct the experiments, one loop-full of biomass of a given species was transferred to an Erlenmeyer flask (100 mL) that contained a 50% TSB broth (20 mL; purchased from Sigma Aldrich, USA). After incubation (24 h, 30 °C), the resulting cellular solution was diluted to 1:50 in cases when the optical density reached a value of 0.1 and combined with a 0.5 mg/mL solution of resazurin (4 mL). This working solution was used for the incubation of the samples.

The tests were carried out using sterile plates with 96 wells (12 columns, each with 8 rows). Two subsequent columns were used for a single tested material, i.e., columns 1 and 2 were used for the first sample, columns 3 and 4 for the second, etc. The first column was inoculated with the working solution (TSB + microorganisms + resazurin, 200 μL) and then suspensions of the tested materials were added (50 μL), starting with the highest concentration at the top row and proceeding to the lowest concentration in the bottom (8th) row. The second column was used for abiotic control and included TSB with resazurin not inoculated with microorganisms (200 μL) and the suspension at the corresponding concentration (50 μL). Ultimately, the studied concentrations ranged from 0.01 to 20 g/L. Biotic control samples were prepared in a separate plate, which included the working solution (250 μL) without the tested materials. Upon preparation of three respective replications, the plates were placed on a rotary shaker and left for incubation (24 h, 30 °C). Afterwards, a visual inspection of the samples was used to establish the characteristic parameters of antimicrobial activity—MIC and MBC (in case of bacteria) or MFC (in case of fungi).

A different approach was used for cement admixtures, which were studied in the form of solid samples (a 5 mm × 5 mm × 2 mm block). In this case, the materials were placed on a sterile Petri plates containing TSB, and then incubated (24 h, 30 °C). A total of three plates were prepared for a single material to ensure a sufficient number of repetitions. Afterwards, each plate was analyzed with respect to the presence of microbial biomass and the extent of microbial growth.

## 3. Results and Discussion

### 3.1. Characteristics of Powders

#### 3.1.1. Crystal Structure

Considering the application properties of zinc oxide and oxide-based materials, special attention should be paid to the crystal structure parameters. Figure 2 shows the XRD patterns for the fabricated ZnO.

In this study, the crystal structure of hexagonal wurtzite (space group *P6*_3_*mc* no. 186; databased card no. 2300112) was recorded as a reference sample. For the obtained samples, the following crystallographic planes were determined: (100), (002), (101), (102), (110), (103), (200), (112), (201), (004) and (202). Interestingly, for the diffraction peaks, especially at 2θ = 31.6, 34.3 and 36.1 degrees, the influence of the synthesis method on the FWHM parameter was noted. Accordingly, the use of conventional heating (hydrothermal route) was associated with a narrowing of the peak base, which may indicate the higher crystallinity of the ZnO-H sample. This may be in line with the previously presented scientific literature. Among others, Wahab et al. [32] found that a reflection of the growth of the fabricated zinc oxide products along the c-axis, indicative of crystallite growth, is the higher intensity and narrower spectral width of the peaks. Table 3 shows the average crystallite size and lattice parameters to better understand the effect of the synthesis method.

The main reason for the differences in the obtained crystallite size parameters can be found in the heating method utilized in the two investigated techniques. In the microwave process, unlike the classic hydrothermal route, heat is generated inside the material as the reactants absorb electromagnetic energy by volume and convert it into heat [33]. Herein, the rate enhancement of the reaction may have a positive effect on the formation of single nanocrystalline particles because of the reduction in the activation energy barrier [34], as we pointed out in our earlier work [22]. The authors of the work [21,35] also proved the good agreement of the refined lattice parameters for the ZnO samples.

#### 3.1.2. Morphology and Textural Properties

The results obtained by SEM and TEM microscopy are shown in Figure 3. For the ZnO produced by the hydrothermal route, a single hexagonal particle shape was observed. The particles had even and smooth edges, and various sizes—generally, lengths of 3.2–5 µm (±0.2 µm) and widths of 1–1.4 µm (±0.1 µm). In addition, interconnections between ZnO monocrystals are visible, which are the result of the in situ synthesis technique used. This was probably influenced by the high constant speed and rapid growth rate. The ZnO-M sample, presented in Figure 3c,d, is characterized by particles with dimensions 300–500 nm long and 30–50 nm wide, indicating their nanobelt shape.

In the case of the sample obtained by the microwave route, particle combination, not due to agglomeration, but due to rapid growth rate, can be noted. This can be partially explained by the effect of applying microwave radiation. In addition, the high reaction rate, as well as the presence of polycrystals in both the hydrothermal and microwave methods, may indicate that the in situ synthesis of ZnO crystals leads to the preparation of particles characterized by epitaxial growth. These observations are consistent with the presented XRD data, which indicate the nanometric size of the crystallites (about 20–40 nm), while the obtained zinc oxide crystals are much larger, which also confirms the epitaxial growth and the obtaining of ZnO polycrystals.

Both the hydrothermal and microwave pathways are widely used in the synthesis of ZnO, as shown by the review of the available scientific literature presented by Kołodziejczak-Radzimska and Jesionowski [10], as well as by Wojnarowicz et al. [33]. In most of the works, however, ZnO is obtained through a precipitation reaction and post-treatment. The described method of synthesis has many disadvantages—in the first place, it induces the agglomeration of particles. In contrast, in the hydrothermal and microwave pathways, both the crystalline growth and the formation of nanoparticles themselves take place directly during the processing. In addition, it should be noted that urea was used as a factor enabling in situ synthesis. Urea decomposes into ammonia during heating (which is a precursor of the precipitation reaction) and carbon dioxide (which causes a slight increase in pressure that is necessary to obtain crystalline materials on the hydrothermal/microwave path) [36]. Hence, both techniques are a further stage in the development of materials science, as they are effective in situ forms of synthesis (in particular, the microwave path), as indicated by Thostenson et al. [37].

When comparing the results we obtained with the available scientific literature, attention should be paid to the similar morphological structures as revealed, among others, by Dac Dien [38], who used a hydrothermal method to obtain hexagonal zinc oxide particles. Moreover, the change in shape and size of the particles of zinc oxide subjected to microwave treatment may, according to Baruah and Dutta [24], be associated with faster growth rates. In addition, Krishnakumar et al. [21] using 10 min of microwave radiation obtained zinc oxide with flower-like structures. The materials used by this group of researchers included hydrazine hydrate, zinc nitrate and ammonia solution, while Hu et al. [39] reported that a sonochemical and microwave method made it possible to obtain linked single-crystalline ZnO rods.

Taking into account the significant differences in the morphologies of materials formed by hydrothermal and microwave techniques, an analysis of textural properties was performed to determine the BET surface area, as well as the total volume and diameter of the pores. The determined results are summarized in Table 4.

The course of nitrogen adsorption/desorption isotherms for the synthesized samples is similar. Both the zinc oxide obtained by the microwave method and that obtained by the hydrothermal method have similar values of BET surface area (2 and 8 m^2^/g) and pore volume (0.005 and 0.020 cm^3^/g). In addition, note the low BET area values. For the ZnO-H sample, the obtained area was equal to 2 m^2^/g, while the pore parameters were at 0.005 cm^3^/g (*V_p_*) and 12.1 nm (*S_p_*), respectively. Obtaining a low surface area for a sample synthesized by hydrothermal pathway should be expected, due to the smooth edges of the generated ZnO crystals observed in the SEM image. In the case of the ZnO-M material, an increase in the BET surface area to the value of 8 m^2^/g in relation to the previously discussed sample was noted, as well as an increase in the volume and diameter of pores to 0.020 cm^3^/g and 24.2 nm, respectively. In the presented research, we saw a clear increase in the surface area of the microwave-assisted material by several times, compared to the identical sample obtained by applying the hydrothermal method, although it can be expected that microwave crystallized materials may exhibit a lower surface area compared to the same sample synthesized using conventional methods [40,41]. Nevertheless, there are numerous literature reports on the microwave synthesis of crystalline zeolites showing significant development of the surface area [42]. Therefore, further research aimed at understanding the nature of the influence of microwave-assisted processes on the BET surface area is necessary, in particular, regarding the formation of the textural properties.

The research presented in our work is related to the use of two methods of ZnO synthesis widely described in the literature—hydrothermal and microwave techniques. However, to the best of our knowledge, none of the work to date has performed in situ synthesis of ZnO via the hydrothermal or microwave pathway. An additional value is the comparison of materials obtained with both methods. The obtained data clearly confirm that, despite similar process conditions, microwave radiation has a crucial impact on the obtained morphology and textural properties.

#### 3.1.3. Thermal Stability

Another of the analyses performed for the synthesized materials was the evaluation of their thermal stability, the results of which are shown in Figure 4 in the form of TGA/DTG curves.

For the obtained ZnO samples, the TGA and DTG curves took different courses. In the case of the material formed by the hydrothermal treatment, there was one weight loss (by approx. 3%) at a temperature of approx. 250 °C. This is associated with the removal of the chemically bound water. In contrast, a total weight loss of nearly 10% was noted for the ZnO-M material. Here, the indicated weight loss is related to the removal of the physically bound water (150 °C) and, subsequently, to the removal of the chemically bound water (250 °C). Additionally, the small peak visible at temperature 430 °C can be attributed to the decomposition of residual organic matter (acetate groups) [43,44,45,46].

Differences in thermal stability of the discussed materials are mainly due to the diverse amounts of bound water (physically and chemically), which also translates into the content of hydroxyl groups on the surface. This outcome has a direct impact on the development of the surface area. Therefore, a higher mass loss for the material with a larger BET surface area should be indicated [47]. On this basis, it can be confirmed that the obtained TGA data correspond well with the previously presented BET data.

#### 3.1.4. Optical Properties

Due to the semiconductor properties of zinc oxide, and to determine the absorption spectrum of materials in order to select an appropriate LED light source, a study of diffuse reflectance spectroscopy was carried out (Figure 5).

The UV absorption for the obtained zinc oxide samples is confirmed by a single band in the range of 400 to 250 nm. Based on the data obtained, we hold that absorption in the UV light range (λ = 395 nm) characterized all samples. This is confirmed by the Kubelka–Munk plot as a function of photon energy, where intermediate band gaps of 3.1 eV are present for all materials regardless of the synthesis method. Furthermore, the previously published energy breaks of zinc oxide samples [45,48,49] agree with to those found for ZnO.

#### 3.1.5. Photocatalytic Activity

The photocatalytic activity was evaluated based on the removal of 4-chlorophenol (20 mg/L). For the photocatalytic tests, a light source best suited to the characteristics of the DRS spectra was selected. Hence, a 50 W UV-LED lamp with a wavelength of λ = 395 nm was applied in the photo-oxidation processes. The obtained photodegradation curves are presented in Figure 6.

The model organic contaminant in the photocatalytic assay was 4-chlorophenol, which shows its toxic effects even at low concentrations (a few μg/L) [50,51]. The authors of papers [21,52] have shown that some pesticides cause cancer diseases in rodents. Compounds that can cause this condition include chlorophenols and phenoxyacetic acid herbicides. Therefore, taking into account the threat posed by the presence of 4-chlorophenol in the environment, it was this compound that was selected to evaluate the photoactivity of the synthesized ZnO.

First of all, it should be noted that the ZnO samples generated by hydrothermal and microwave methods were characterized by a high removal of 4-chlorophenol. In both cases, nearly 100% degradation efficiency of the tested pollutant was gained after 180 min, while satisfactory results were obtained after approx. 90 min. After this time, approx. 85% and 75% efficiency of 4-chlorophenol removal was noted for the ZnO-M and ZnO-H samples, respectively. To confirm the high photo-oxidative activity of the obtained products, photolysis tests were carried out. These showed that 4-chlorophenol is slightly degraded under the influence of UV-LED light alone, amounting to approx. 10%. Hence, the results of the photocatalysis confirm that the obtained ZnO samples were characterized by high photoactivity in the process of photocatalytic removal of the tested pollutant.

As it has been stated for the photocatalytic degradation of 4-CP in aqueous titanium dioxide suspensions, this reaction can be formally described by the Langmuir–Hinshelwood kinetics model [53]:(1)$$-\frac{dC}{dt}=\frac{kK_{c}}{1+K_{c}}$$
where *dC/dt* is the rate of degradation, *k* is the apparent reaction rate constant, and *K_c_* is the adsorption coefficient of the substance to be degraded. At the considered initial concentration (*C*_0_ = 10 mg L^−1^), the course of the plot in Figure 7 indicates that the 4-CP decay obeys a pseudo-first-order kinetic law (Equation (2)):(2)$$-\frac{dC}{dt}=kC\;or\;C_{t}=C_{0}e^{-kt}$$

Considering the high photoactivity of the synthesized materials we obtained, in the next stage of our research, an attempt was made to suggest the supposed mechanism of photodegradation. Based on the available scientific literature, the expected mechanism was indicated as a free radical reaction induced by UV light [54,55]. As a result of excitation, photo-generated electrons (e^−^) are created in the conduction band (CV). This leads to a photo-reduction process that produces, e.g., hydroxyl radicals. The oxidation process, caused by holes (h^+^) in the valence band (VB), leads to the formation of hydroxyl radicals. The oxidation reaction involves surface hydroxyl groups or adsorbed water.

Based on the work of Stafford et al. [56], our study confirms that the presence of the hydroxyl radical (*OH) is a crucial parameter influencing the photo-oxidation of the tested contaminant. Hence, this is the primary oxidant. Moreover, our work confirms that of previous researchers who found that holes (h^+^) in the valence bond are responsible for the degradation process, albeit to a lesser extent [53,57]. Therefore, our research indicates that the main degradation route is the hydroquinone pathway.

According to the presented pathway, hydroquinone is mainly oxidized to 1,2,4-benzenetriol, and the main reactions in the photo-oxidation process of 4-chlorophenol are as follows: hydroxylation, dihydroxylation, hydrations, and decarboxylation [58].

The World Health Organization (WHO) has recognized phenolic compounds, e.g., chloro- and nitrophenols and bis-phenols, as pollutants with negative environmental effects. Given this, many researchers are working on the reduction of phenol derivatives. Table 5 compares the photo-oxidative removal of 4-chlorophenol using different photocatalysts.

According to the presented review on the photodegradation of 4-chlorophenol, it should be noted that various photocatalysts are used for the removal of this contaminant, among other oxides, such as g-C_3_N_4_, TiO_2_ and ZnO, and plasmonic materials. The results of degradation efficiency obtained by us were better than or similar to those reported in other studies.

In our research, we used an innovative “green” alternative to UV light–narrow-spectrum light-emitting diodes. These, according to our experience so far, constitute an innovative approach compared to the commonly used mercury and xenon lamps. Apart from the harmfulness to the environment related to the disposal of mercury, in the era of sustainable development, and given the aim to limit the consumption of electricity, it is extremely important to use innovative technologies that can successfully replace the non-ecological processes used today. Bearing this in mind, both the zinc oxide obtained by us and the constructed light source seem interesting subjects for further research, and may be the basis for developing LED photocatalysis.

#### 3.1.6. Antimicrobial Properties

The results of antimicrobial tests carried out for ZnO-M and ZnO-H powders are presented in Table 6. There is a clear difference in terms of the growth-inhibiting effects of both ZnO powders, which indicates that the method of their preparation has a profound influence on this property. The ZnO-M powder samples exhibited notably higher antimicrobial activity; as in most cases, the MIC values were at least two times lower compared to the values obtained for ZnO-H. Furthermore, the MBC value could be determined for some of the studied microorganisms, whereas in the case of ZnO-H samples, this parameter always exceeded the studied range of concentrations. It should also be emphasized that the inhibiting effect of the studied samples was species-specific, as diverging MIC and MBC values were obtained for different bacterial species. Overall, the *C. albicans* yeast species was characterized by the highest resistance (MIC at 20 mg/L) towards the tested powders, whereas *E. coli* was most susceptible to their antimicrobial action (MIC at 0.63 mg/L; MBC at 2.5 mg/L). It should be noted that there was no visible trend in terms of the impact of the studied powders on either Gram-positive (*P. putida, P. aeruginosa, E. coli*) or Gram-negative bacterial species (*B. cereus, S. aureus*). Based on the obtained results, the following order of tolerance of the microbial species (from highest to lowest) to the tested compounds may be established: *C. albicans* > *S. aureus* = *P. aeruginosa* > *P. putida* > *B. cereus* > *E. coli*.

### 3.2. Cement Composites Analysis

#### 3.2.1. Cement Mortars Setting Time

As part of the research into cement mortars amended with zinc oxides, the beginning of setting time was analyzed, the results of which are presented in Table 7. The analyzed reference sample is a pure mortar without admixture, for which the beginning of the setting time was determined as 170 min. As compared to zinc oxide-free mortars, mortars containing zinc oxides are characterized by a longer hydration period. For the ZnO-H sample, this time was 600 min, and was 550 min for ZnO-M. This represents a 430 and 380 min delay in initial setting time, respectively.

The prolonged beginning of the initial setting time for the clinkerization process for materials doped with zinc oxide was also demonstrated in the work of Bordoloi et al. [64]. Moreover, in the work of [65], the authors used zinc oxide and alccofine to create cement composites containing both of these components. Here, the addition of alccofine was constant (10%), while the zinc oxide was introduced in various proportions in the amounts of 0%, 0.25%, 0.5%, 0.75% and 1.0%. The research results presented by the authors clearly indicate that with the increase in the ZnO content in the mortar, the beginning of the setting time increases.

#### 3.2.2. Plasticity of Cement Mortars

As part of the test carried out on the spread of fresh cement mortar, the plasticities of the cement mortar of the reference sample and mortars containing 0.1 wt.% of ZnO-H and ZnO-M admixture were compared and presented in Figure 8. The obtained results of the analyses clearly indicate no negative influence of admixtures on the plasticity of the cement mortar. The reference sample reached a flow of 18.0 cm, as did the mortar containing the ZnO-H admixture. A slightly higher spread, 18.5 cm (which is about 3% higher), was achieved for the ZnO-M-doped mortar. A slight increase in plasticity may be related to the specificity of the admixture used, because the ZnO-M material, as shown in Table 3, is characterized by smaller particles, and the achieved greater plasticity may be the result of better material dispersion in the mortar.

A study of the plasticity of cement mortar was also carried out by the authors of the work [66], who converted 2%, 4% and 6% of cement to nano-ZnO and bentonite. Another analyzed material was nano-Al_2_O_3_, which replaced 1%, 2% and 3% of the cement. In the work, they achieved spreads in the range of 150–200 mm. For nano-ZnO, no changes in the flow rate were demonstrated. The authors of [5] conducted a study of flow on mortars containing nano-SiO_2_, nano-TiO_2_, nano-Al_2_O_3_ and nano-ZnO admixtures. In their research, they showed that the addition of nano-SiO_2_ and nano-Al_2_O_3_ significantly reduced the liquidity of the mortar. The tests were carried out on samples containing 0.5, 1.0, 1.5, 2.0 and 3.0 g nano-SiO_2_ and 0.25, 0.5, 0.75 and 1.0 g nano-Al_2_O_3_. Compared to the discussed nanomaterials, the addition of nano-TiO_2_ and nano-ZnO had a slight effect on the plasticity of the mortar.

In a related work by Liu et al. [5], mortars with an admixture of 0.5, 1.0, 2.0 and 3.0 g of nano-TiO_2_ and 0.5, 1.0, 1.5 and 2.0 g of nano-ZnO were analyzed. The resulting observations indicated the influence of the BET surface area of the investigated nanomaterial on the plasticity properties of the mortar. Here, nano-oxides with a high BET surface area, such as SiO_2_ and Al_2_O_3_, showed a decrease in plasticity, while materials with a smaller BET surface area, such as nano-ZnO and nano-TiO_2_, did not have such a significant effect.

#### 3.2.3. Flexural and Compressive Strength Tests

The results of the strength tests after 1, 7 and 28 days for both flexural and compression are presented in Figure 9.

The flexural strength (see Figure 9a) for the mortar without admixtures is characterized by the relatively highest values, both after 7 and 28 days, in relation to mortars amended with zinc oxides. However, after the first day, the highest value of flexural strength was achieved by a sample of ZnO-H-doped mortar. The lowest value, after 1, 7 and 28 days of curing, was achieved by the mortar with ZnO-M admixture.

The presented results of compressive strength (see Figure 9b) for the reference sample, after 1, 7 and 28 days, are, respectively, equal to 27.7, 49.7 and 63.1 MPa. Despite the slightly better flexural strength, the compressive strength values for this mortar are the least favorable. Mortar doped with zinc oxide obtained by the hydrothermal method (ZnO-H) was characterized by relatively high early strength (after 1 day, the value of 30.1 MPa was reached, and after 7 days, 57.4 MPa—these figures represent increases of ~9% and ~15.5%, respectively) despite the extended initial setting time of the mortar. The compressive strength value after 28 days was slightly lower than the highest value achieved in the presented tests, and amounted to 71.3 MPa (an increase of ~13% compared to the reference sample). The compressive strength values for ZnO-M mortar were characterized by the lowest early strength of 27.1 MPa after 1 day (a decrease of ~2%), as well as a significant increase in the further stage of seasoning; after 7 days, the value of 57.2 MPa was reached, which is a result better by 13%, compared to pure mortar. Ultimately, this system reached the highest compressive strength value after 28 days, equal to 71.5 MPa, which is higher by 12% when compared to the reference sample.

Flexural strength tests for pastes containing an admixture of synthesized zinc oxide obtained by the precipitation method were also carried out by Arefi and Rezaei-Zarchi [67]. In their research, they used an admixture of 0.05%, 0.1%, 0.2%, 0.5% and 1.0% of zinc oxide and assessed its effect on flexural strength after 7, 14, 21 and 28 days. The obtained test results clearly indicated that 0.5% of the dopant was the most advantageous amount of zinc oxide used, for which the obtained values of flexural strength were the highest. With the 1.0% admixture, a decrease in value was observed, which may be due to the presence of too many nanoparticles that could combine with the released lime during the hydration process.

The authors of the work [68] performed compressive strength tests for mortars containing commercially available zinc oxide admixture. The test was carried out after 3, 7, 14 and 28 days of maturation for samples containing an admixture of zinc oxide in the amounts of 0.5%, 1.0%, 1.5% and 2%. The obtained and presented test results clearly indicate that that the most favorable values of compressive strength were obtained for the mortar containing 0.5% ZnO admixture. Despite the low strengths in the initial hydration period, the strength gain after 28 days was very significant.

The introduction of an admixture of zinc nano-oxide has a significant impact on the behavior of the cement matrix. Zinc oxide nanoparticles affect the hydration process in the early stage of the reaction, and therefore the initial strength obtained for cements containing ZnO admixture may be lower compared to the reference sample of pure cement. This value may slightly increase for the strength tested after 7 days, achieving the most significant increases after 28 days of curing. The strength increases evident after 7 days are due to the filler effect of ZnO nanoparticles (by which the number of pores is reduced), which reduces the permeability of the cement matrix, as well as increasing the density and compressive strength [69].

#### 3.2.4. Microstructural Analysis Results

The microstructure study was performed for cement mortars after 28 days of maturing, and the obtained SEM photos are presented in Figure 10. The reference sample shown in Figure 10a,a’ is characterized by a relatively uniform structure. Micropores and needle-like ettringite are visible. Additionally, the presence of hexagonal flakes of large molecules derived from calcium hydroxide can be observed. The microstructure of the mortar containing the ZnO-H admixture, visible in Figure 10b,b’, is characterized by a more compact structure, with sparse air bubbles. It can be observed that this material is characterized by a greater number of structures visible on the surface, which may confirm the beneficial effect of the used admixture. Visibly, it increases the number of active centres associated with the cement hydration process. Figure 10c,c’ shows the SEM pictures of a cement mortar containing ZnO-M dopant. The presented microstructure of the material, as in the case of ZnO-H admixture, is characterized by a greater amount of cement hydration products, compared to pure mortar. Interestingly, the obtained crystals are characterized by a smaller sizes compared to those for the ZnO-H mortar. This outcome may be related to the specificity of the admixture used, because the zinc oxide obtained by the hydrothermal method is characterized by a hexagonal shape, with a length greater than the particle diameter. In the case of the zinc oxide obtained by the microwave method, the particles are smaller in size, but tend to aggregate and agglomerate. However, due to the smaller size of ZnO-M particles, a greater number of potential active centers are obtained after their introduction into the cement mortar, which results in a more homogenous C-S-H phase in the entire sample volume.

The confirmation of a more even distribution of ZnO-M particles in the cement matrix, compared to the mortar containing ZnO-H admixture, is visible in the SEM photos shown in Figure 11 and the EDX mapping for the mortars discussed in this study. Figure 11a,a’’ show the images obtained for a zinc oxide-amended mortar obtained by a hydrothermal method, and Figure 11b,b’’ for zinc oxide obtained by a microwave method. In comparing the mapping of the doped mortars presented in Figure 11a’,b’, a larger area can be observed for the composite containing ZnO-M. This is also confirmed by Figure 11a’’,b’’, in which only the image of identification of ZnO in the sample is presented. The ZnO-M-doped mortar is characterized by a greater homogeneity of the sample, and ZnO is distributed throughout the sample volume. In the case of the analyzed ZnO-H composite sample, places where zinc oxide is absent can be observed.

In their work, Liu et al. [5] performed an analysis of the microstructure of mortars containing an admixture of, among others, nano-ZnO. Their observations for the mortar containing 2.0 wt.% of nano-ZnO admixture revealed the presence of numerous unhydrated cement particles, Aft and C-S-H gel. The samples presented in the article, due to the high content of zinc oxide, are characterized by a visible isolation of the C-S-H gel groups and a large number of needle-hydrates on the surface of the cement particles. A consequence of the large and accelerated increase in cement hydration caused by the high amount of dopant may be a faster loss of strength by this material.

When analyzing the microstructures of mortars doped with nanomaterials, one can often observe the accumulation of the action of nanoparticles, which accelerate the pozzolanic reaction in the cement matrix. Another reaction that develops is the formation of a C-S-H gel layer around unreacted binder grain. This gel compacts the entire structure and binds all particles, and is the main product of hydration, being the main factor influencing the strength of the material. In addition, the ettringite formed during the cement hydration process can cause the growth of thin needles, creating increasingly heavier rods that thicken and additionally cross-link the entire structure. These effects demonstrate the nuclei effect held by nano-particles. This allows for a uniform structure via the better distribution of C-S-H gel [66].

#### 3.2.5. Antimicrobial Properties of Cement Composites

The results of antimicrobial tests carried out with the use of cement composites with ZnO-H and ZnO-M are presented in Table 8 and Figure 12. Our investigation of the antimicrobial effects of cement composites confirmed the results previously described for powder samples of the obtained zinc oxides. The composite material amended with ZnO-M efficiently inhibited the growth of microorganisms at the agar/sample interface, since not even a single colony could be observed (Figure 12b). In the case of the composite with ZnO-H, there were several colonies present on the right side of the sample (Figure 12c, the contrast has been adjusted for better visibility of the biomass). Nevertheless, the most extensive microbial growth was observed in the case of pure cement (which was used as a reference sample) (Figure 12a, biomass growing in all directions below the sample), which demonstrates that the presence of ZnO-H contributes to a marginal inhibitory effect. The combined results obtained for powder samples and cement composites indicate that the method of obtaining ZnO has a notable impact on its antimicrobial properties.

The antimicrobial properties of zinc oxide nanoparticles are reflected by their various applications (e.g., as components of food packaging, self-cleaning materials and accelerating agents for improved wound healing). ZnO may inhibit the growth of microorganisms via direct interaction (e.g., release of Zn^2+^ ions) or through indirect mechanisms (e.g., generation of reactive oxygen species) [70].

It should be emphasized that the actual ability of zinc oxide nanoparticles to inhibit the growth of specific strains may notably differ [71]. The exact effect is influenced by the particle size, morphology, concentration and the presence of defects [72], which overall corresponds to the method of their preparation. This statement is supported by the results presented in the framework of this study, as contrasting antimicrobial effects were observed for ZnO-H and ZnO-M in the forms of both powders and cement composites.

While some reports indicate that the inhibitory effect of zinc oxide is higher in the case of Gram-positive bacteria due to the differences associated with their cell wall structure [73], this was not observed in the framework of this study. In contrast, our work indicated that ZnO-M powder samples efficiently inhibited the growth of Gram-positive *B. cereus*, with the highest antimicrobial effect herein being observed in the case of Gram-negative *E. coli*. Other representatives of Gram-positive and Gram-negative bacterial species, however, exhibited higher resistance to the zinc oxide, which was at a similar level. The high efficiency of ZnO against *E. coli* that we noted was confirmed in other studies [74], which imply that the effect may result from the disruption of cellular membrane integrity and the subsequent destabilization of transport processes. In our work, the MIC values we obtained correspond well with those of other studies, as minimal inhibitory concentrations up to 40 mg/L have been reported for ZnO nanoparticles [75].

In order to further enhance the antimicrobial effect of cement composites, it is possible to include several strategies, such as the use of a second oxide (e.g., CuO for greater antifungal effect), surface modification, or the addition of photosensitizers for increased generation of ROS [76]. Still, perhaps the most fundamental issue is to change the synthesis method in order to achieve ZnO with satisfactory structural parameters that may contribute to high antimicrobial activity. The results presented in this study strongly support the above-mentioned concept, as substantially different effects of zinc oxides obtained via different methods were observed.

Future studies should, therefore, focus on exploring the relationship between the structural properties of ZnO and its antimicrobial activity in order to establish the best parameters (i.e., size, shape, etc.), and on providing adequate synthesis methods of materials with optimal morphology.

## 4. Conclusions

The obtained zinc oxides were characterized by different morphologies and microstructures. The crystal structure assessed by XRD analysis confirmed the crystallographic planes of hexagonal wurtzite structure. The application of scanning (SEM) and transmission electron microscopy (TEM) allowed for determining that the particles obtained by the hydrothermal method are single hexagonal shapes with even and smooth edges, whereas nanobelt shapes were noted for the particles obtained by the microwave method. The textural properties tests we carried out confirmed the relatively low BET surface area values for ZnO-H (2 m^2^/g) and for ZnO-M (8 m^2^/g) samples, while the thermogravimetric analysis of the obtained zinc oxides revealed significant differences between the products obtained.

For the obtained oxides, our assessment of optical properties undertaken via DRS analysis confirmed similar absorption properties in the UV light range (λ = 395 nm). With regard to their photocatalytic activity in terms of the removal of 4-chlorophenol, both obtained oxides turned out to be very effective, with almost 100% efficiency after 180 min.

In the final stage of testing, we investigated their antibacterial properties against selected species of Gram-negative bacteria, Gram-positive bacteria, and a single fungus. Our work confirmed that ZnO-M particles are characterized by significantly higher antimicrobial activity. Indeed, in many cases, the MIC values were at least two times lower than for ZnO-H. Additionally, we were able to evaluate the MBC value (which was not possible for ZnO-H particles). Based on the obtained results, the tolerance of microbial species to the tested compounds was established (highest to lowest) as: *C. albicans* > *S. aureus* = *P. aeruginosa* > *P. putida* > *B. cereus* > *E. coli*.

Our work indicates that zinc oxide has a delaying effect on the cement hydration process. Here, the specific initial setting time of the mortar for composites containing ZnO-H was 600 min, while that for ZnO-M was 550 min, and that of the reference sample was 170 min, while for fresh cement mortar, the spread rate was also determined.

Our study did not show that the zinc oxide admixture had a negative effect upon the plasticity of the mixture, as we reached values identical to the reference sample (ZnO-H-doped composite, 18 cm) or slightly higher (mortar with ZnO-M, 18.5 cm—an increase of about 3%). The doped samples were, however, characterized by more favorable compressive strength values (the highest value herein being obtained for the composite with ZnO-M—an increase of about 12%). Additionally, we performed surface mapping using EDX analysis, which allowed us to visualize the distribution of zinc oxide particles in the cement matrix. The analysis of the microstructure of cement composites confirmed the beneficial effects of zinc oxides in increasing the cement hydration products visible in the SEM photos.

## Figures and Tables

**Figure 1 materials-15-01069-f001:**
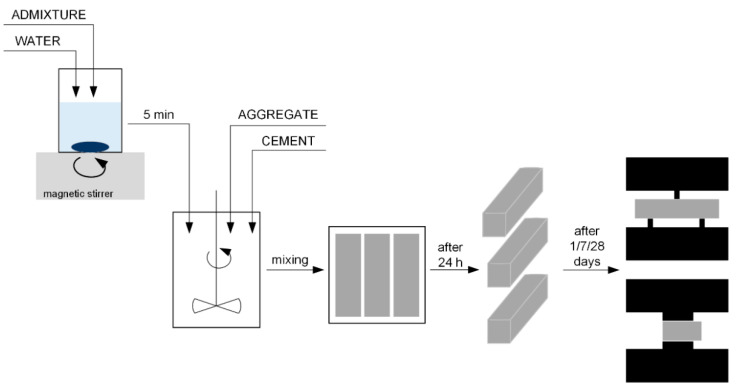
Schematic diagram for the preparation of cement composites.

**Figure 2 materials-15-01069-f002:**
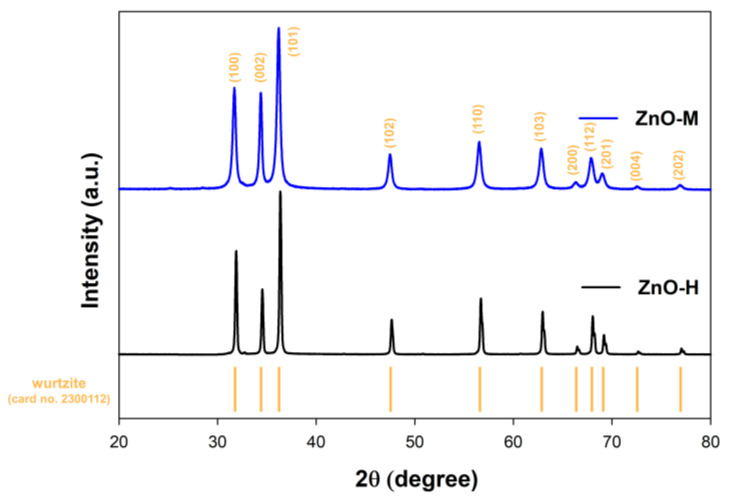
The XRD patterns for synthesized ZnO materials.

**Figure 3 materials-15-01069-f003:**
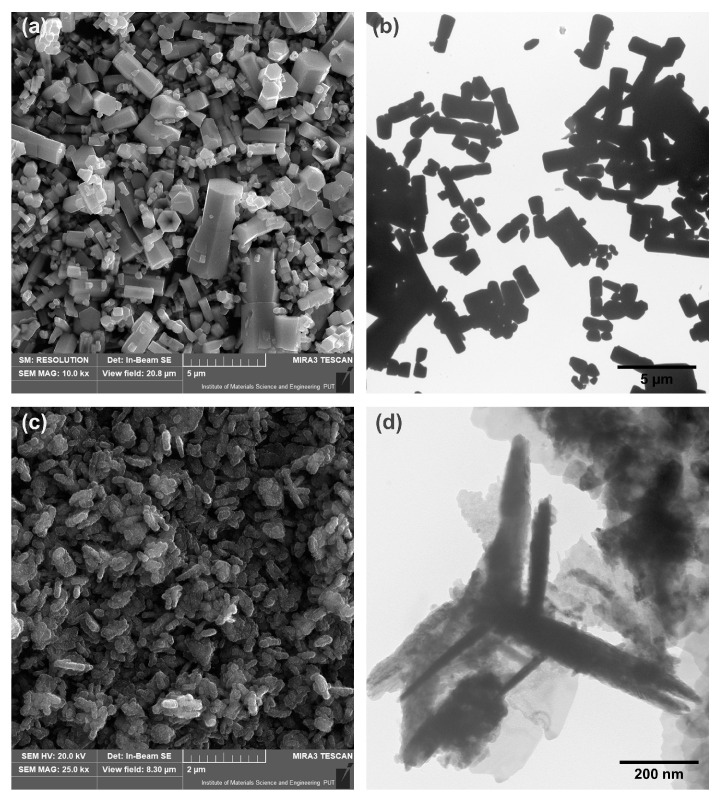
The SEM and TEM images for ZnO-H (**a**,**b**) and ZnO-M (**c**,**d**).

**Figure 4 materials-15-01069-f004:**
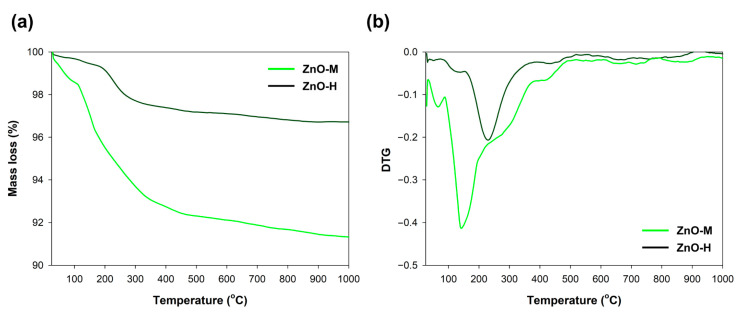
Thermogravimetric (TGA) (**a**) and derivative thermogravimetric (DTG) (**b**) curves of ZnO samples.

**Figure 5 materials-15-01069-f005:**
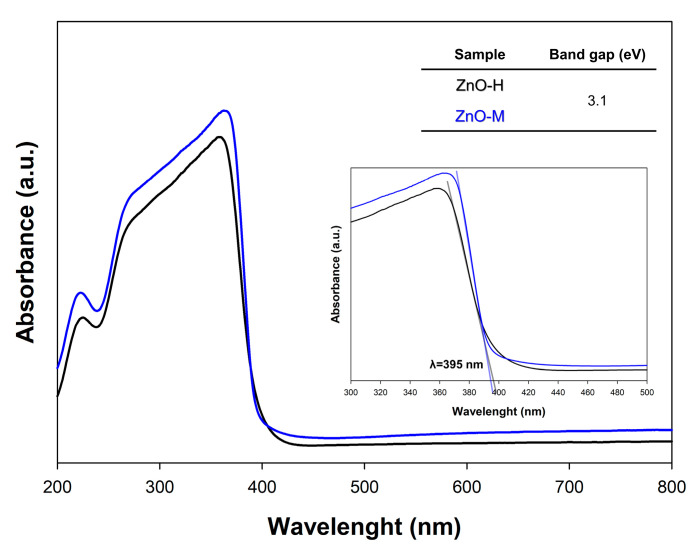
The DRS spectra for ZnO materials.

**Figure 6 materials-15-01069-f006:**
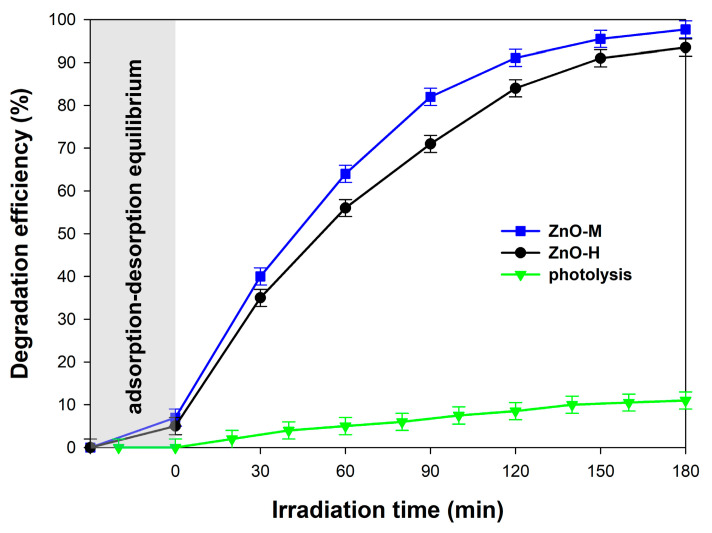
Effect of the photo-oxidation process carried out for synthesized ZnO in the presence of 4-chlorophenol.

**Figure 7 materials-15-01069-f007:**
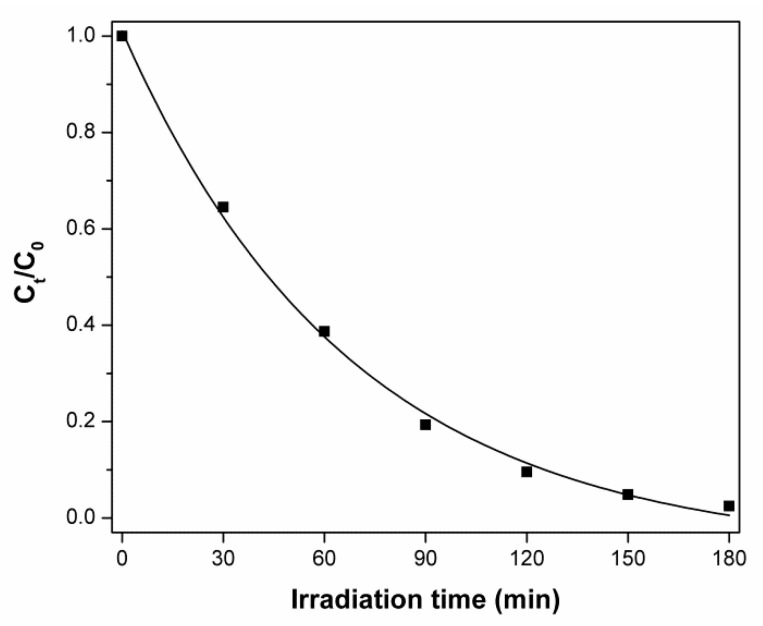
Normalized 4-CP concentration as a function of irradiation time in the presence of ZnO-M sample. Initial concentration of 4-CP − *C*_0_ = 10 mg/L.

**Figure 8 materials-15-01069-f008:**
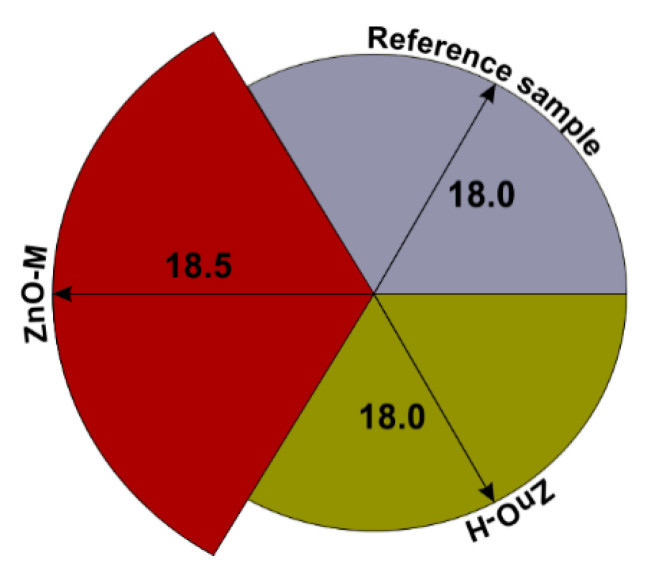
Plasticity of cement mortars analyzed in this study.

**Figure 9 materials-15-01069-f009:**
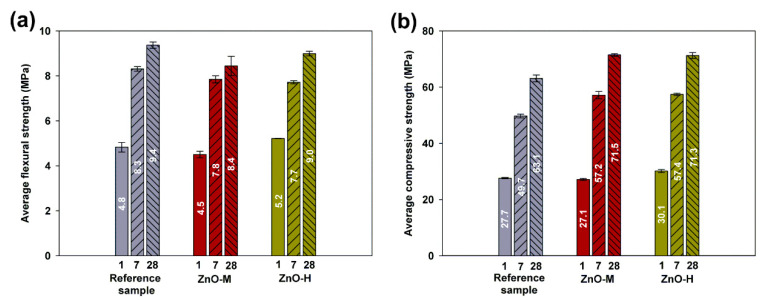
Flexural (**a**) and compressive (**b**) strength test results for reference sample and mortars doped with ZnO-M and ZnO-H.

**Figure 10 materials-15-01069-f010:**
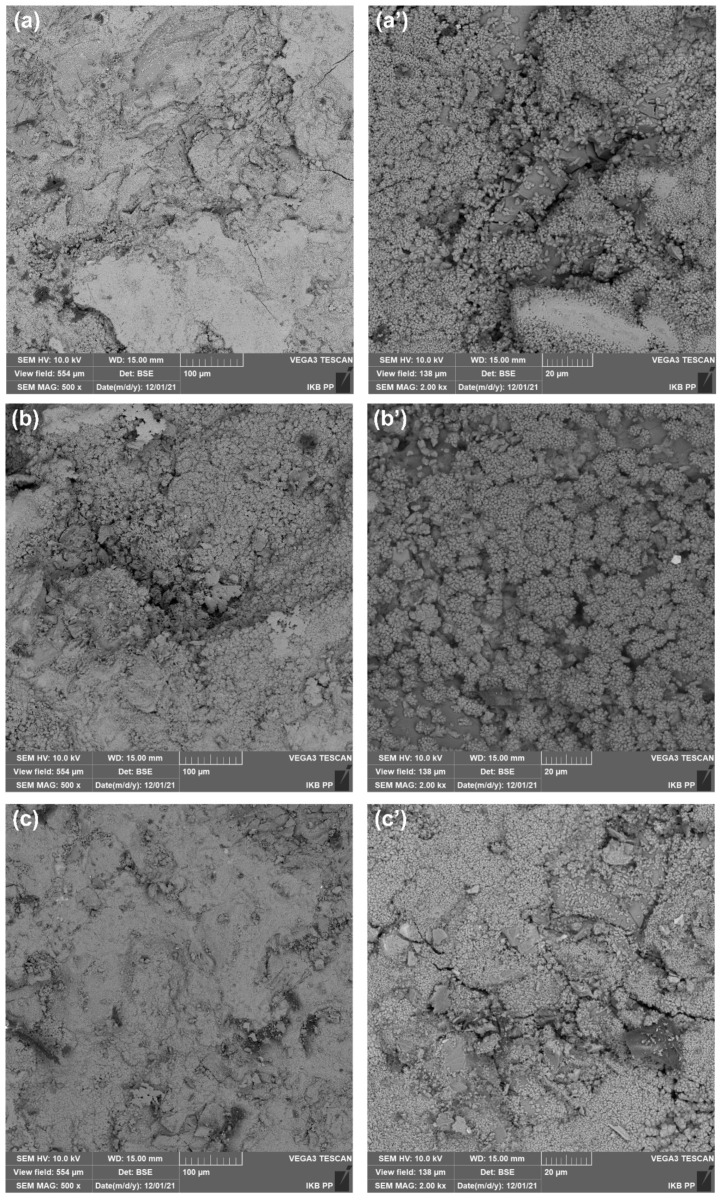
SEM images of (**a**,**a’**) reference sample and cement composites doped with (**b**,**b’**) ZnO-H and (**c**,**c’**) ZnO-M in different magnifications.

**Figure 11 materials-15-01069-f011:**
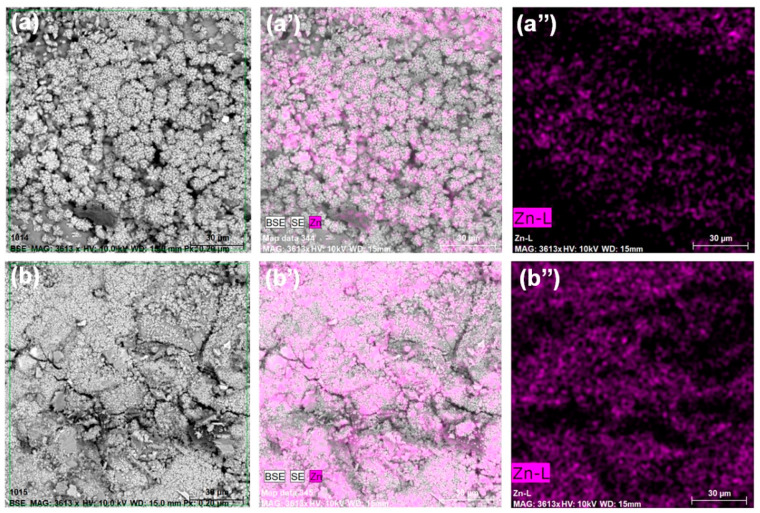
SEM–EDS analysis of cement composites doped with ZnO-H (**a**–**a’’**) and ZnO-M (**b**–**b’’**).

**Figure 12 materials-15-01069-f012:**
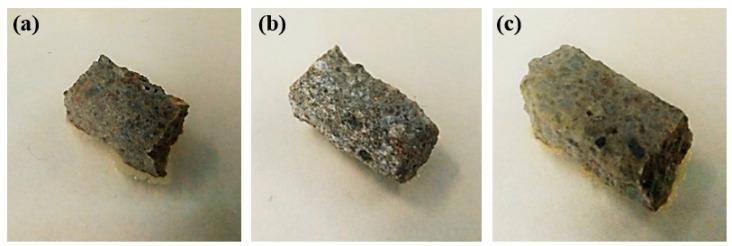
Example images of cement composite samples after 24 h in Petri dishes: pure cement (**a**), cement composite with ZnO-M (**b**) and cement composite with ZnO-H (**c**).

**Table 1 materials-15-01069-t001:** Conditions of ZnO synthesis using hydrothermal and microwave pathways.

Synthesis Method	Process Conditions
hydrothermal	T = 160 °C T = 6 h
microwave	T = 160 °C t = 1 min * P = 300 W

* Time for the microwave method is the holding time at 160 °C.

**Table 2 materials-15-01069-t002:** Composition of the analyzed cement mixtures.

Sample	Cement (g)	Aggregate (g)	Water (mL)	Admixture (g)
CEM I	450	1350	225	-
ZnO-H				0.45
ZnO-M				0.45

**Table 3 materials-15-01069-t003:** The average size of crystallites and lattice parameters of synthesized ZnO materials.

Sample	The Average Size of Crystallites (nm)	Lattice Parameters (Å)	
		a	c
ZnO-H	48.5 (±0.2)	3.24408	5.19691
ZnO-M	27.3 (±0.3)	3.25611	5.21687

**Table 4 materials-15-01069-t004:** The result of textural properties of ZnO materials.

Sample	*A_BET_* (m^2^/g)	*V_p_* (cm^3^/g)	*S_p_* (nm)
ZnO-H	2	0.005	12.1
ZnO-M	8	0.020	24.2

**Table 5 materials-15-01069-t005:** Comparison of removal of 4-chlorophenol using different photocatalysts.

Material	Degradation Conditions			Degradation Efficiency	Ref.
	Concentration	Amount of Photocatalyst	Type of Light Source		
ZnO-M	20 mg/dm^3^	0.1 g/dm^3^	UV-LED (50 W)	90% (in 120 min)	This work
ZnO-rGO	5 mg/dm^3^	0.1 g/dm^3^	Hg–Xe lamp (200 W)	94% (in 60 min)	[59]
Ag–AgI/Fe_3_O_4_@SiO_2_	10 mg/dm^3^	0.1 g/dm^3^	Hg–Xe lamp (250 W)	90% (in 30 min)	[60]
ZnO-TiO_2_	25 mg/dm^3^	0.1 g/dm^3^	Xe-lamp UV-B (16 W)	95% (in 80 min)	[61]
ZnO	50 mg/dm^3^	0.2 g/dm^3^	Xe-lamp UV-A (6 W)	95% (in 180 min)	[62]
ZnO/g-C_3_N_4_	42 mg/dm^3^	0.1 g/dm^3^	Hg–Xe lamp (150 W)	80% (in 360 min)	[63]

**Table 6 materials-15-01069-t006:** Antimicrobial test results obtained for ZnO-M and ZnO-H powders.

Compounds	*Pseudomonas putida* *G−*		*Pseudomonas aeruginosa* *G−*		*Bacillus cereus* *G+*		*Escherichia coli* *G−*		*Staphylococus aureus* *G+*		*Candida albicans* *Yeast*	
	MIC *	MBC *	MIC *	MBC *	MIC *	MBC *	MIC *	MBC *	MIC *	MBC *	MIC *	MFC *
ZnO-M	10	20	10	Above 20	5	20	0.63	2.5	10	Above 20	20	Above 20
ZnO-H	20	Above 20	20	Above 20	20	Above 20	Above 20	Above 20	20	Above 20	Above 20	Above 20

* all data presented in mg/L.

**Table 7 materials-15-01069-t007:** Initial setting time obtained for reference samples and cement mortars doped with ZnO.

Cement Mortar	Initial Setting Time (min)
Reference sample	170
Mortar doped with ZnO-H	600
Mortar doped with ZnO-M	550

**Table 8 materials-15-01069-t008:** Evaluation of microbial growth in the presence of pure cement and cement composites with ZnO-H and ZnO-M.

Sample	Microbial Growth
Pure cement (reference)	+++ (extensive growth)
Cement composite with ZnO-H	++ (high growth)
Cement composite with ZnO-M	− (lack of growth)

## Data Availability

The data presented in this study are available on request from the corresponding author.

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
