# Peer review of "The In Situ Hydrothermal and Microwave Syntheses of Zinc Oxides for Functional Cement Composites"

_materials, 2022, doi:10.3390/ma15031069_

Round 1

Reviewer 1 Report

Kindly find the attached comments

Author Response

Dear Reviewer, 

Thank You for Your insightful review of our work, which contributed to a better understanding of the scientific problems related to the subject of the publication and will help with the elimination of potential errors in the future.We would also like to express our gratitude for the revision of our manuscript and the opportunity to re-submit it, incorporating all of the Referees’ suggestions. Our comments and changes are noted below, and are marked in yellow in the manuscript. The authors reported The in situ Hydrothermal and Microwave Syntheses of Zinc Oxides Towards Obtaining Functional Cement Composites Overall, the work may be of interest to a broader audience. However, the authors should address the following points outlined below to improve scientific quality. After the suggested revisions are carefully addressed, this work may be considered for publication.

Query 1: I suggest modifying the as “The in situ Hydrothermal and Microwave Syntheses of Zinc Oxides for Functional Cement Composites”

Answer 1: We thank the Reviewer for this suggestion. The changes have been made in the manuscript and are marked in yellow.

Query 2: The abstract is too lengthy it should clear and informative with important highlights. The author mentioned two times on the other hand in abstract section. Which is confusing.

Answer 2: We thank the Reviewer for this suggestion. Selected abstract sentences have been rewritten and are marked in yellow.   

Query 3: In the background section, Line 60 sentence ZnO is used e.g. in such sectors as pharmaceutical and cosmetic industries as a component sentence is not clear.

Answer 3: We thank the Reviewer for this suggestion. The sentence has been rewritten and is marked in yellow. 

Query 4: From the title it looks the main highlight of this work is Hydrothermal and Microwave Syntheses of ZnO however author have not provided any literature review and importance of these 2 methods in introduction section

Answer 4: We thank the Reviewer for this comment. Following the recommendation of the Reviewer, but also similar suggestions from other Reviewers, a relevant literature background was added to the introduction section.  

Query 5: In SEM and TEM results Line 247 to 249 the author mentioned about agglomeration of particles. The author should explain the reasons for agglomeration

Answer 5: We sincerely thank the reviewer for this question. The main reason for agglomeration for samples synthesized with the microwave method is due to the high kinetics of the reaction. According to the available literature, it is known that the microwave method enables the synthesis of nanomaterials. However, the high process rate in many cases leads to particle aggregation, as we also mentioned in our earlier work [1]. [1]     A. Kubiak, S. Å»óÅ‚towska, E. GabaÅ‚a, M. Szybowicz, K. SiwiÅ„ska-Ciesielczyk, T. Jesionowski, Controlled microwave-assisted and pH-affected growth of ZnO structures and their photocatalytic performance, Powder Technol. 386 (2021) 221–235. https://doi.org/10.1016/j.powtec.2021.03.051 

Query 6: In same Line 248-the author mentioned it’s difficult to measure the single particle size. However, the author mentioned approximate size ranges were 300–500 nm for 249 length and 30–50 nm for width. Is this the aprox size of single particle? If yes then delete the prvious sentence ‘ size of a single particle is difficult to determine; IF not then clearyfy the sentence about this size single particle or multiple?

Answer 6: We thank the reviewer for this valuable suggestion. Appropriate changes have been made to the manuscript. 

Query 7: Can you compare your SEM/TEM results with other research work ?Answer 7: We thank the reviewer for this valuable comment. The necessary changes have been made to the manuscript. 

Query 8: The used the However many times throughout the manuscript for example Line 253, 261, 263, 265

Answer 8: We would like to thank the Reviewer for this valuable remark. The submitted manuscript has been carefully checked; consequently, grammatical and typographical errors were removed. 

Query 9: Are the results reproducible. Kindly show data with % of reproducibility?

Answer 9: Many thanks to the reviewer for this question. First of all, we want to point out that the method of synthesis presented by us was fully reproducible and characterized by high efficiency. It is a simple in situ precipitation method in a hydrothermal / microwave reactor, while the process efficiency itself can be described as approximately 90%.               On the other hand, the photo-oxidation tests performed by us were carried out in accordance with scientific standards. Each of the spectrophotometric measurements was repeated several times, while the test error was determined from the standard deviation. Moreover, we would like to point out that the test error was also included in the results presented in Figure 6. 

Query 10: There are many grammatical errors, recheck the manuscript once again for all typo errors.

Answer 10: We thank the Reviewer for this comment. The whole manuscript has been carefully checked with regard to editorial and language issues. 

Query 11: In introduction section Cite reference  M. W Alam et al "Study to Investigate the Potential of Combined Extract of Leaves and Seeds of Moringa oleifera in Groundwater Purification." Int. J. Environ. Res. Public Health 2020, 17, 7468; doi:10.3390/ijerph17207468.

Answer 11: We thank the Reviewer for this suggestion. The article has been cited in the manuscript and is marked in yellow. 

The whole manuscript has also been carefully checked with regard to editorial and language issues.

We look forward to hearing from you.

Yours faithfully,

Prof. Agnieszka Åšlosarczyk (corresponding author),

Poznan University of Technology

Reviewer 2 Report

The manuscript lacks novelty as well as in-depth discussion about the ZnO's properties.  What's new here? Only two different synthesis methods are reported. As compared to current scientific progress I feel this work could have been better presented. This manuscript needs thorough modification.

  1. The structural study of ZnO's is missing. Authors need to calculate the Crystallite size, lattice parameters of the two different ZnO's
  2. The photocatalysis section lacks the kinetics study.
  3. The language needs substantial improvement, please revise the language carefully.
  4. The response of reported ZnO is lower than compared materials. Elucidate why?
  5. The photocatalytic mechanism is not described.
  6. BET surface area is reported but there is no graph of BET N2 adsorption-desorptions.
  7. The conclusion is too broad, it should be concise.
  8. It would be better to examine products of the photocatalytic reaction
  9. Quantitative information on the experimental results should be added in the abstract.
  10. A quenching/Scavenger experiment should be conducted to confirm which radicals have participated in this photocatalytic reaction system.

Author Response

Dear Reviewer, 

Thank You for Your insightful review of our work, which contributed to a better understanding of the scientific problems related to the subject of the publication and will help with the elimination of potential errors in the future.We would also like to express our gratitude for the revision of our manuscript and the opportunity to re-submit it, incorporating all of the Referees’ suggestions. Our comments and changes are noted below, and are marked in yellow in the manuscript. The manuscript lacks novelty as well as in-depth discussion about the ZnO's properties. What's new here? Only two different synthesis methods are reported. As compared to current scientific progress I feel this work could have been better presented. This manuscript needs thorough modification.

Query 1: The structural study of ZnO's is missing. Authors need to calculate the Crystallite size, lattice parameters of the two different ZnO's

Answer 1: We thank the reviewer for this valuable suggestion. However, we would like to point out that both the crystallite size and the lattice parameters are listed in Table 3. 

Query 2: The photocatalysis section lacks the kinetics study.

Answer 2: We thank the reviewer for this valuable comment. The necessary changes have been made to the manuscript. 

Query 3: The language needs substantial improvement, please revise the language carefully.

Answer 3: We thank the Reviewer for this comment. The whole manuscript has been carefully checked with regard to editorial and language issues.  

Query 4: The response of reported ZnO is lower than compared materials. Elucidate why?

Answer 4: Many thanks to the Reviewer for this question. We would like to draw the Reviewer's attention that in the presented comparison of the efficiency of the 4-chlorophenol photodegradation process, the ZnO samples obtained by us were compared with two- or three-component materials. Hence, a higher efficiency of this type of materials seems to be expected due to a different photo-oxidation mechanism, e.g. a hereto coupling of type II or scheme Z etc. However, despite only one photoactive component, the obtained photocatalyst was characterized by a similar performance to the materials mentioned. This may be related to the effective design of the entire photocatalytic system, and therefore not only the photocatalyst itself, but also the light source, which, to the best of our knowledge, is not found in the scientific literature. In many of the presented works, previous researchers focussed exclusively on photoactive material using conventional light sources (mercury or xenon lamps), which, apart from the fact of high price and the content of harmful elements, are also highly energy-consuming. Hence, despite slightly lowered performance parameters of the degradation process, our system is characterized by a longer service life and lower energy consumption. Therefore, in our opinion, this is the right path in the development of photocatalytic processes. 

Query 5: The photocatalytic mechanism is not described.

Answer 5: We thank the Reviewer for this comment. Based on scientific knowledge, the free radical process should be indicated as the main mechanism of the photo-oxidation process [2]. As a result of excitation, photo-generated electrons (e-) are formed in the conduction band (CV), which leads to the photo-reduction process that generates reactive oxygen species, e.g. hydroxyl radicals. On the other hand, holes (h+) in the valence band (VB) result in the oxidation process by reacting with adsorbed water or surface hydroxyl groups, resulting in the formation of hydroxyl radicals. 

[2]     B. Ohtani, Titania photocatalysis beyond recombination: A critical review, Catalysts. 3 (2013) 942–953. https://doi.org/10.3390/catal3040942.

 Query 6: BET surface area is reported but there is no graph of BET N2 adsorption-desorptions.

Answer 6: We thank the Reviewer for this comment. We agree with the Reviewer that isotherm of N2 adsorption-desorption would be an interesting analysis enriching the comprehensive physicochemical characteristics of the synthesized materials. Unfortunately, our scientific team does not have direct access to this analytical technique, but definitely, we will try to include this aspect in future research. 

Query 7: The conclusion is too broad, it should be concise.

Answer 7: We thank the Reviewer for this suggestion. As recommended by the Reviewer, but also by other Reviewers, the conclusions were shortened. 

Query 8: It would be better to examine products of the photocatalytic reaction.

Answer 8: We thank the reviewer for this valuable remark. According to the available literature, the route of degradation of 4-chlorophenol is the hydroquinone pathway [3]. Hydroquinone is mainly oxidized to 1,2,4-benzenetriol, and most of the ring cleavage comes from this compound. However, a modest amount of tetraol and two additional acyclic six-carbon compounds can also be observed. The authors also indicated that the presented photo-oxidation pathway is suchlike to other aromatic compounds, including quinoline, naphthalene, and pyridine, whereas, after opening the ring, further decomposition into shorter organic compounds occurs, and consequently their mineralization. Finally, the main reactions in the photocatalytic removal of 4-chlorophenol are proposed as follows: hydroxylation, dihydroxylation, hydrations and decarboxylation. 

[3]     X. Li, J.W. Cubbage, T.A. Tetzlaff, W.S. Jenks, Photocatalytic degradation of 4-chlorophenol. 1. The hydroquinone pathway, J. Org. Chem. 64 (1999) 8509–8524. https://doi.org/10.1021/jo990820y.

Query 9: Quantitative information on the experimental results should be added in the abstract.

Answer 9: We thank the Reviewer for this suggestion. The changes have been made in the abstract and are marked in yellow. 

Query 10: A quenching/Scavenger experiment should be conducted to confirm which radicals have participated in this photocatalytic reaction system.

Answer 10: We thank the Reviewer for this suggestion. We agree with the Reviewer that a quenching/scavenger experiment would be an interesting analysis enriching the comprehensive photocatalysts characteristics of the synthesized ZnO samples. Unfortunately, our scientific team does not have direct access to this analytical method, but definitely, we will try to include this aspect in future research.

The whole manuscript has also been carefully checked with regard to editorial and language issues.

We look forward to hearing from you.

Yours faithfully,

Prof. Agnieszka Åšlosarczyk (corresponding author),

Poznan University of Technology

Reviewer 3 Report

- The manuscript requires English proof-reading. There are some grammatical errors and vague sentences that should be revised before publication. Authors should especially focus on revising abstract and conclusions which are the main parts of a paper.- For the Abstract section, it is the opinion of this reviewer to rephrase the section and focus on the main purpose of the topic and the mainly outcomes conducted from the experimental tests. No need for sub-introduction. 

- Check Figure 9. SEM images should show some cracks and other occurrences.- For the conclusion section. It is the opinion of this reviewer to rephrase the section and focus on the main outcomes.- Make sure that the bibliography style meets the journal standards.

Author Response

Dear Reviewer, 

Thank You for Your insightful review of our work, which contributed to a better understanding of the scientific problems related to the subject of the publication and will help with the elimination of potential errors in the future.We would also like to express our gratitude for the revision of our manuscript and the opportunity to re-submit it, incorporating all of the Referees’ suggestions. Our comments and changes are noted below, and are marked in yellow in the manuscript. 

Query 1: The manuscript requires English proof-reading. There are some grammatical errors and vague sentences that should be revised before publication. Authors should especially focus on revising abstract and conclusions which are the main parts of a paper.

Answer 1: We thank the Reviewer for this comment. The whole manuscript has been carefully checked with regard to editorial and language issues. The Authors also focused attention on revising abstract and conclusions. Once again we thank the Reviewers for this suggestion.  

Query 2: For the Abstract section, it is the opinion of this reviewer to rephrase the section and focus on the main purpose of the topic and the mainly outcomes conducted from the experimental tests. No need for sub-introduction.

Answer 2: We thank the Reviewer for this suggestion. Selected abstract sentences have been rewritten and marked in yellow.   

Query 3: Check Figure 9. SEM images should show some cracks and other occurrences.

Answer 3: We thank the Reviewer for this comment. In the presented SEM images there is no visible cracks because samples used for this analysis was additionally formed. We used the material that remained after the creation of the cement bars. All samples were matured at the same time under the same conditions. In addition, the mortars demonstrated a much more compact microstructure and good adhesion at the aggregate-cement paste interface, where micro-cracks or air pores can often be found. In the mortars that were additionally thickened with nano-sized additives, a very homogeneous structure was obtained. Furthermore, in the cases analyzed, for the most part, the microstructure was evaluated at high magnifications, resulting in the absence of visible micro-cracks. Once again we thank the Reviewer for this suggestion. 

Query 4: For the conclusion section. It is the opinion of this reviewer to rephrase the section and focus on the main outcomes.

Answer 4: We thank the Reviewer for this suggestion. As recommended by the Reviewer, but also by other Reviewers, the conclusions were shortened. 

Query 5: Make sure that the bibliography style meets the journal standards.

Answer 5: We thank the Reviewer for this comment. The bibliography has been checked and all changes made are highlighted in yellow. 

The whole manuscript has also been carefully checked with regard to editorial and language issues.

We look forward to hearing from you.

Yours faithfully,

Prof. Agnieszka Åšlosarczyk (corresponding author),

Poznan University of Technology

Reviewer 4 Report

Dear Editor-in-Chief: Materials (ISSN 1996-1944)

Dear Ms. Amy Hao

MDPI Materials Editorial Office

Paper The in situ Hydrothermal and Microwave Syntheses of Zinc Ox-2 ides Towards Obtaining Functional Cement Composites. The paper is worth publishing however, the following concerns should be addressed in revising the manuscript:

Abstract

1- It requires clarification that this idea is new.

2- Please add an overview of the practical investigation

Methodology

1- It is preferable to add the most important results in numbers or percentages

Introduction

1- Adding a special section on the scientific background of the research.

2- It is preferable to clarify more about what is new in this study (materials or methods)?

 Materials and Methods

 1- Inclusion of a collection of references and standard specifications for all tests

2- The methodology is unclear, preferably more detailed (Antimicrobial Properties of Pristine Admixtures and Cement Composites)

Results and discussion

1- The paper should be linked to each other better and supported by previous research.

Conclusion

1- It is preferable to add the most important results that support this research.

2- The most important search results should be displayed as on points for readability

thank you

Author Response

Dear Reviewer, 

Thank You for Your insightful review of our work, which contributed to a better understanding of the scientific problems related to the subject of the publication and will help with the elimination of potential errors in the future.We would also like to express our gratitude for the revision of our manuscript and the opportunity to re-submit it, incorporating all of the Referees’ suggestions. Our comments and changes are noted below, and are marked in yellow in the manuscript. Paper The in situ Hydrothermal and Microwave Syntheses of Zinc Ox-2 ides Towards Obtaining Functional Cement Composites. The paper is worth publishing however, the following concerns should be addressed in revising the manuscript:

Query 1: Abstract1- It requires clarification that this idea is new.2- Please add an overview of the practical investigation

Answer 1: We thank the Reviewer for this suggestion. The text has been supplemented with the Reviewer's suggestions and any changes have been highlighted in yellow. 

Query 2: Methodology1- It is preferable to add the most important results in numbers or percentages

Answer 2: We thank the reviewer for this valuable remark. Relevant data have been added to the manuscript and highlighted in yellow.

 Query 3: Introduction1- Adding a special section on the scientific background of the research.2- It is preferable to clarify more about what is new in this study (materials or methods)?

Answer 3: We thank the Reviewer for this comment. Following the recommendation of the Reviewer, but also similar suggestions from other Reviewers, a relevant literature background was added to the introduction section.  

Query 4: Materials and Methods 1- Inclusion of a collection of references and standard specifications for all tests2- The methodology is unclear, preferably more detailed (Antimicrobial Properties of Pristine Admixtures and Cement Composites)

Answer 4: We thank the Reviewer for this comment. Pertinent changes have been made in manuscript and are marked in yellow. 

Query 5: Results and discussion1- The paper should be linked to each other better and supported by previous research.

Answer 5: We thank the Reviewer for this suggestion. We have discussed some of the results previously, but in addition, as suggested by the Reviewer, some parts of the results have been modified and discussed in more detail, based on the available literature. 

Query 6: Conclusion1- It is preferable to add the most important results that support this research.2- The most important search results should be displayed as on points for readability

Answer 6: We thank the Reviewer for this suggestion. As recommended by the Reviewer, but also by other Reviewers, the conclusions were shortened. 

The whole manuscript has also been carefully checked with regard to editorial and language issues.

We look forward to hearing from you.

Yours faithfully,

Prof. Agnieszka Åšlosarczyk (corresponding author),

Poznan University of Technology

Round 2

Reviewer 1 Report

The authors responded to all the comments from me and I am satisfied with the revision.

Author Response

Dear Reviewer, 

Thank You for Your insightful review of our work, which contributed to a better understanding of the scientific problems related to the subject of the publication and will help with the elimination of potential errors in the future.We would also like to express our gratitude for the revision of our manuscript and the opportunity to re-submit it, incorporating all of the Referees’ suggestions. Our changes are marked in yellow and green in the article.

We look forward to hearing from you.

Yours faithfully,

Prof. Agnieszka Åšlosarczyk (corresponding author),

Poznan University of Technology

Reviewer 2 Report

The authors have tried their best to revise the manuscript. However, some important clarifications are still missing e.g. 1. Kinetics study is added in the revised manuscript but authors did not provide its graph; 2. The Photocatalytic study is discussed however its mechanism did not provide even after being asked in revision, only published research reference is provided. Authors should discuss, draw their own mechanism ; 3. The scavenger study is not difficult to study, it's just the addition of different scavengers, i.e., holes, electrons, hydroxyl radicals etc. in the reaction and further examination of their effect on the photocatalytic reaction. According to this study, the authors would have studied the real photocatalytic mechanism. 

Author Response

Dear Reviewer, Thank You for Your insightful review of our work, which contributed to a better understanding of the scientific problems related to the subject of the publication and will help with the elimination of potential errors in the future.We would also like to express our gratitude for the revision of our manuscript and the opportunity to re-submit it, incorporating all of the Referees’ suggestions. Our comments and changes are noted below. The authors have tried their best to revise the manuscript. However, some important clarifications are still missing.Querry 1: Kinetics study is added in the revised manuscript but authors did not provide its graph; Answer 1: Thank you very much for the valuable remark. We have already added Figure 7 (see below) indicating that the course of photocatalytic degradation of 4-chlorophenol in the presence of ZnO sample occurs under a pseudo-first-order kinetic law. Querry 2: The Photocatalytic study is discussed however its mechanism did not provide even after being asked in revision, only published research reference is provided. Authors should discuss, draw their own mechanism ; Querry 3: The scavenger study is not difficult to study, it's just the addition of different scavengers, i.e., holes, electrons, hydroxyl radicals etc. in the reaction and further examination of their effect on the photocatalytic reaction. According to this study, the authors would have studied the real photocatalytic mechanism. Answer 2 and 3: Dear Reviewer, thank you very much for your valuable suggestions. However, we would like to draw your attention to a few additional points. First of all, we would like to emphasize that the main assumption of the presented work was the synthesis of new ZnO samples and the cement composites obtained on their basis. Therefore, the work presents a comprehensive physicochemical characteristics of both powder materials and the aforementioned composites. We consider this aspect to be a scientific novelty, which was emphasized in the part introduction. On the other hand, we know how extremely important it is to search for innovative applications for synthesized materials/composites, which is why we decided to present a multifunctional approach using both photoactivity and biocidal properties. Bearing in mind the obtained results, and thus the occurrence of both photocatalytic (for powders) and antibacterial (for composites) activity, in the paper we presented possible mechanisms of these activities, based on the available scientific literature. Of course, we fully agree with the Reviewer's suggestion that the proposed research aimed at a detailed description of the mechanism is extremely important, and for photocatalysis, it can be considered crucial. Therefore, we will focus on those aspects in future research that will be devoted exclusively to the approach to the mechanism, using already well-characterized material. Such an approach seems justified to us because a comprehensive physicochemical characterization is the key to further considerations on the mechanism. Hence, we would like to thank you once again for your valuable suggestion, which will surely be used by our team in future research. 

We look forward to hearing from you.

Yours faithfully,

Prof. Agnieszka Åšlosarczyk (corresponding author),

Poznan University of Technology
